# A multi-dimensional assessment of the 2021 mouse plague in New South Wales, Australia: Economic impacts and policy responses

Walter Okello[1]*, Kerry Collins[2‡], Aditi Mankad[2‡], Lucy Carter[2‡], Peter R. Brown[3‡]

1 Environment Research Unit, Commonwealth Scientific and Industrial Research Organisation, Canberra, Australia Capital Territory, Australia, 2 Environment Research Unit, Commonwealth Scientific and Industrial Research Organisation, Brisbane, Queensland, Australia, 3 Health & Biosecurity Research Unit, Commonwealth Scientific and Industrial Research Organisation, Canberra, Australia Capital Territory, Australia

☉ These authors contributed equally to this work.
‡ These authors also contributed equally to this work.
* walter.okelo@csiro.au

## Abstract

Despite the periodic mouse plague outbreaks in Australia which largely occur due to favourable climatic conditions, their economic impacts remain understudied. To bridge this knowledge gap, the present study analysed the economic impacts of the 2021 mouse plague in New South Wales (Australia) among households, farms, and businesses/facilities. We further analysed the influence of selected cost parameters on mouse bait rebate claims by the farmers as well as the relative efficiency of the chemical mouse control options used by farmers to minimize avoidable crop yield losses. Our study found that the total direct cost of the 2021 mouse plague was A\$ 100.62 million, with farmers bearing 67.10% of the total cost. It was also revealed that the type of farming influenced the likelihood of a farmer claiming or not claiming mouse bait rebate. The most efficient chemical mouse control option was the combination of anticoagulants used around buildings and zinc phosphide used in pastures and crops as it reduced avoidable crop yield losses more than each rodenticide when used independently. However, more research is required on how other variables may influence the efficiency of mouse control methods to forestall future outbreaks of mouse plagues and their associated economic impacts.

## Introduction

Mouse plague refers to the occurrence of an unusually substantial number of mice infesting a region and causing severe damage particularly in agriculture. Mouse plagues can be considered as disasters given the significant impacts they can have on the economy, the environment and human health [1,2]. In Australia, house mouse (*Mus musculus*) plagues have been known to occur sporadically at five-to-ten-year

**Data availability statement:** All relevant data are within the paper and its Supporting Information files.

**Funding:** This research was funded by the NSW Department of Primary Industries and Regional Development. The funders had no role in study design, data collection and analysis, decision to publish, or preparation of the manuscript.

**Competing interests:** The authors have declared that no competing interests exist.

intervals throughout the agricultural regions especially in the south-eastern parts of the country (News South Wales, Victoria, and South Australia) where broadacre crops are grown [3–5]. However, the economic impacts of mouse plagues on grain growers and rural communities are not well understood.

One of the most severe mouse plagues in the south-eastern region of Australia can be traced back to 1970 where several field crops were extensively damaged [4]. These plagues resulted in considerable economic losses to agricultural production because mice damaged sown seeds and growing crops as well as stored produce [6]. In 1980, there was another mouse plague in the state of Victoria and the Murrumbidgee irrigation area of New South Wales (NSW) and this time mainly affecting irrigated sunflower crops [3,7]. There was another mouse plague in 1984 in Australia that mostly affected the state of Victoria [7]. A decade later (1993), there was yet another severe mouse plague in South Australia and Victoria resulting in an estimated cost of Australian dollars (A$) 60.6 million to grain growers alone [7]. The five to ten-year cycle of severe mouse plagues struck again in 2011, affecting four states namely NSW, Queensland, South Australia, and Victoria. In NSW alone, the Farmers' Federation estimated that 3 million hectares of crops were affected [8]. The latest mouse plague was in 2021 which mostly affected the central and northern parts of NSW following exceptional rainfall after several years of drought [2,9]. This coincided with the COVID-19 pandemic that severely affected many rural and regional NSW communities [2].

Mouse plagues affect households through damage to property, cost of replacing damaged property, and expenditures incurred from buying mouse bait and household cleaning products [7]. Farmers are impacted by mouse plagues through production losses, damage to farm machinery, and mouse bait expenditures. In Australia, the wild mouse causes substantial damage to mainly winter cereals such as wheat and other broadacre crops such as maize, sorghum, legume, and oil seeds [3,10–14]. For example, in the 1993 southern Australia mouse plague, approximately 450,000 hectares of cereal, legume and oilseed crops were affected [7]. Mouse plagues are also known to damage infrastructure in rural areas [7]. The economic impact of mice on crops is majorly through damage to all the stages of crop development by digging up newly planted seeds, cutting tillers to gain access to nutrients contained within the crop and feeding on the developing grain as the crop matures [15]. For livestock farmers, the economic impact of mouse plagues occurs due to damage to hay and fodder crops, destruction of feed grain, and reduction in productivity [16]. For example, [7] reported that the 1993 mouse plague resulted in high piglet mortality rates, increased stress among weaners, and caused a 20% reduction in poultry egg production. At the state level, [7] reported that the cost of the 1980 and 1984 mouse plagues in the state of Victoria were A$ 15–20 million and A$ 10.2 million, respectively. These estimates were based only on crop yield losses. Further, the cost of the 1993 mouse plague to the states of Victoria and South Australia were A$ 14.4 million and A$ 46.2 million, respectively [7]. In contrast to previous cost estimates that majorly focussed on crop damages, the cost estimations done in 1993 included other cost parameters such cost of mouse baiting, resowing, and cost of mouse plagues to businesses/facilities.

Because of the high economic impacts of mouse plagues, many Australian farmers use chemical methods, especially baiting with anticoagulants and zinc phosphide [13,17,18]. Anticoagulants are mostly used around buildings while zinc phosphide is used in pastures and cropping [17]. Moreover, broadscale application of zinc phosphide in cropping has been found to be cheap and effective with a very low (less than 1% yield loss) economic injury level, i.e., the density of the pest population at which the economic damage equals the cost of control measures [15,19]. To minimize the social and economic impact of the 2021 mouse plague, the NSW government initiated A$ 150 million mouse bait rebate scheme. Of the A$150 million public expenditure, households, small businesses, and farmers applied to receive up to A$500 each, A$ 500 each, A$ 1,000 each, respectively [20].

The current understanding of the economic impact of mouse plagues in Australia is based on data collected some 30 years ago from the 1993 mouse plague report [7,16]. Thus, the majority of the information on the economic impact of mouse plagues are found in grey literature in the form of reports resulting in limited knowledge on the economic impact of pests to the agricultural sector and ultimately inefficiency in resource allocation. Moreover, it is important to have a standard approach for assessing the economic impact of mouse plagues or similar natural disasters to enable trend analysis and comparison between and within states and countries. Apart from understanding the economic cost of mice, it is important to understand how cost data can be used to inform policy. It should be noted that although cost analysis is important in understanding the opportunity cost of preventing mouse plague, its contribution to economic decision making is limited [21].

The aim of the study was to understand the economic impact of the 2021 mouse plague and model the cost data to further gain practical insights into how mouse plagues interventions can be made more efficient by policymakers. Specifically, the study had three broad objectives. The first objective was to assess the cost of the 2021 mouse plague in NSW and compare it with the past economic impact of mouse plagues in Australia. The second objective was to identify the factors that influenced the mouse bait rebate scheme, the policy instrument of interest. The third objective was to compare the efficiency of the type-specific chemical mouse bait used by the farmers during the mouse plague to minimize crop yield losses. Therefore, this study went beyond the standard cost analysis and provided insights on how cost data can be utilised for decision making.

## Materials and methods

### Study site

The study site for this research was the grain-growing area of regional NSW (west of the Great Dividing Range) in Australia. In NSW, winter crops include cereals such as wheat, barley, oats, and triticale; oilseeds such as canola and safflower; and pulses such as lupin, chickpea, faba bean and field pea. Summer crops include grain sorghum, rice, and cotton. Forage and fodder crops include forage sorghum, Pennisetum, millet, lablab, cowpeas, soybeans, grain sorghum and maize [22]. According to the Australian Bureau of Statistics (ABS), the value of crops produced in NSW in 2021/22 was A$ 18 billion [23].

The total area of crops grown in NSW was estimated to be 1,622,496 hectares in 2021 with a total of 23,901 farms of which the majority (28.7%) were specialised beef cattle farming [23,24]. Other major types of farms included specialised grain-sheep or grain-beef cattle farming (15%), specialised sheep farming (13.3%), other grain growing farms (11.4%), sheep-beef cattle farming (8.7%), dairy farming (2.7%), fruit and tree nut farming (2.7%), outdoor vegetable growing (2.3%), grape growing (1.6%), and horse farming (1.5%) [24]. Between 2021 and 2022, the total average farm cash income was A$ 779,000 [24].

According to the 2021 population census, the population of regional NSW (excludes Sydney) stood at 3,121,679 (38.6% of the total population in NSW) and the total number of businesses in regional NSW was 262,189 which was 32.2% of the total number of businesses in NSW [25]. In NSW, the total average household expenditure in 2021/2022 was A$ 152,119 [26].

## Study design and conceptual framework

The study was designed to first collect and analyse empirical data and then choose the most affected entities for further analysis as part of a multi-analytical approach [27]. Further analysis entailed assessment of the factors that influenced mouse control policy and relative efficiency assessment of the chemical mouse control options used by crop farmers during the outbreak to minimize crop yield losses. The conceptual framework is as shown in Fig 1. The information obtained from the assessment of the cost of the mouse plague as well as the factors affecting mouse control policy and the relative efficiency analysis of mouse control methods employed during mouse plagues can be used to inform mouse control policies and investment plans. It was hypothesized that use of chemical methods such as mouse bait was the predominant approach for managing mice by farmers during mouse plagues as opposed to non-chemical methods such as electric fencing and biological approaches.

## Data collection

The populations of inference were the households, farms, and businesses or facilities in NSW whereas the target populations were the households, farms, and businesses/facilities that were affected by the 2021 mouse plague. The methods used for data collection from the target population are described below.

**Web survey, sampling, and participant recruitment.** A web survey, which is part of the broader online survey methodology, was used for data collection and to ensure a wide coverage (i.e., reduce coverage error) [28–30]. The web survey process was divided into three stages namely pre-survey, survey, and post-survey. Pre-survey stage involved determining the sampling technique, questionnaire development and testing, drafting a non-response strategy, and drafting a general management protocol. The survey stage involved participant recruitment, administration of the

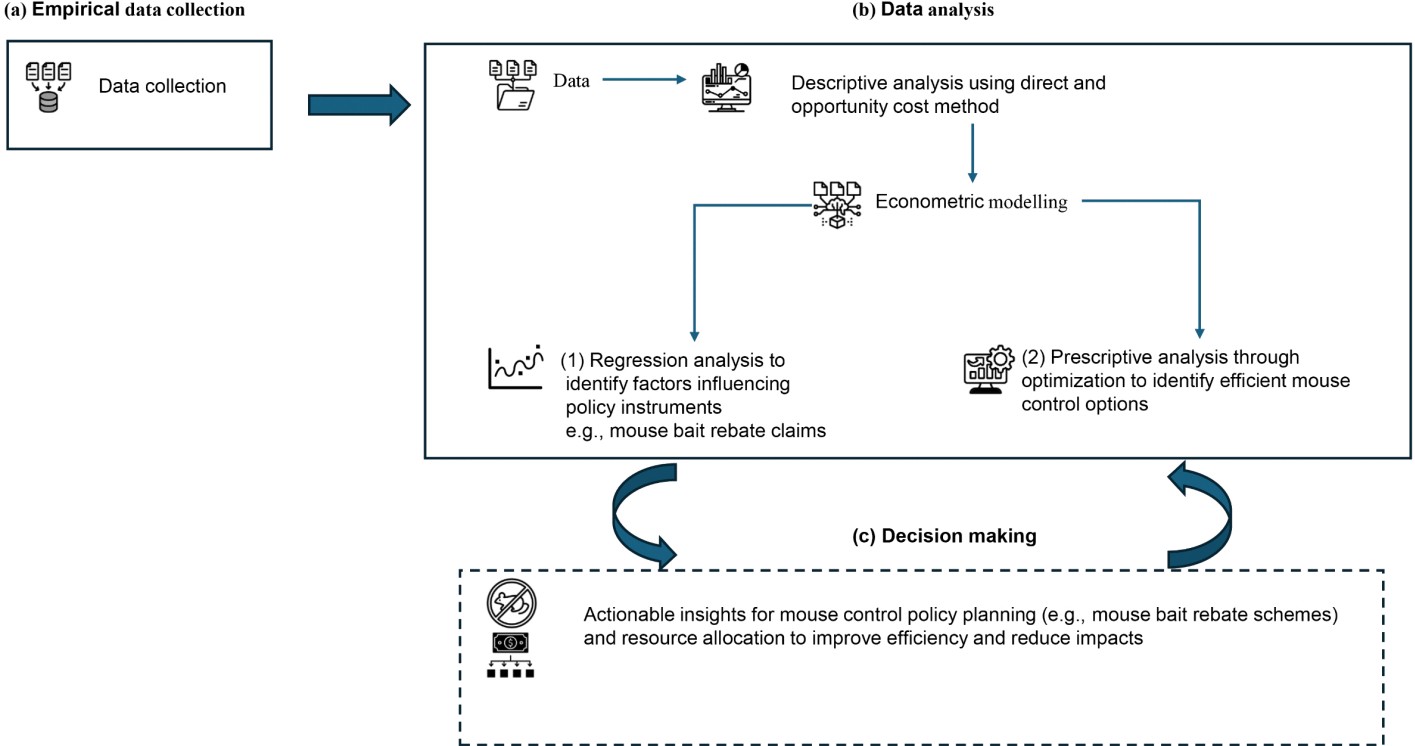

**Fig 1. Conceptual framework of the study.** (a) Empirical data collection (b) Data analysis (c) Decision making.

questionnaire and responding to any concerns raised by the respondents whereas the post-survey stage entailed data preparation and analysis of the preliminary results.

The sampling technique used in the study was the combination of restricted (i.e., probability-based pre-recruited online panel) and unrestricted (random non-panel survey) web survey methods [29]. The unrestricted web survey where a survey link was distributed to potential participants via industry newsletters and social media was used to supplement the online panel to improve representativeness particularly for remote regions, reduce sampling bias, increase diversity and obtain respondents that are fresher (i.e., not used to taking online surveys) [31]. Additionally, the age and gender of the online panel was matched with the ABS to further improve the representativeness of the target population [32].

The survey instrument was a semi-structured questionnaire derived from literature review and was first developed in Microsoft Word and then transferred to a web form. The survey instrument was designed to capture the economic impact of the 2021 mouse plague across households, farms, and businesses/facilities and was divided into five blocks of questions. The first block of questions captured the socio-demographic information such as age, sex, local government area (LGA), and postcode. The second block of questions captured the descriptive household mouse plague impacts (e.g., social, mental, and economic impacts). The third block of questions was dedicated to farm level economic impacts (e.g., value of crop yield losses, value of seeds and seedlings damaged, value of livestock productivity decrease, value of damaged farm equipment, and expenditures on mouse bait) as well as the mouse control methods used and their related cost. The fourth block of questions captured the types of businesses or facilities that were affected by the mouse plague and the economic impact (e.g., income losses, value of damaged goods, expenditures on cleaning supplies, and expenditures on mouse bait). The last block of questions was the description of other types of economic impacts that were experienced by households, farmers, and businesses including their monetary values.

A major consideration during the development of the questionnaire was the cognitive aspect of the response process to reduce mental effort and response bias [33]. This was done in recognition that the data was being collected a few years after the event and was achieved by: 1) ensuring that the questions were easy to comprehend, 2) randomizing the question order, 3) ensuring that most questions were single item (i.e., addressed one dimension of the economic impact at a time), 4) use of radio buttons for closed-ended questions (e.g., press 'YES' or 'NO'), 5) ensuring that the web survey form was interoperable across several types of devices, e.g., desktop, mobile phones, etc., 6) use of nonresponse prompts to encourage participants answer the questions before proceeding, and 7) enabling the participants to save and continue the web survey at any time; an important feature that enabled participants to pause whenever they had mental fatigue or had to find the relevant documents (e.g., farm records and insurance claims) needed to answer the questions. Participants were also able to quit the survey at any time. The survey instrument was then pre-tested using face-to-face interviews and expert evaluation after its development. The survey took an average of 15 minutes to complete.

The survey, which ran from 25 October to 31 December 2023, was programmed and hosted by a reputable online market research provider in consultation with the research team. The provider managed the representative sample recruitment process which involved emailing invitations for survey participation to their national databases targeting those living in the target NSW region as well as collating survey responses collected via the shared survey link embedded within newsletters and social media advertisements. Participants did not receive any monetary incentives to participate in the survey. However, those recruited through the representative panel process received a token incentive from the provider (not the research team) in the form of redeemable membership points.

Participation in the study was divided into three stages namely screening, provision of information and consent terms, and answering of the questions. At the start of the survey, participants were asked if they were 18 years old or over and only those who were 18 years old or over proceeded to the next screening question which checked if their households, farms, or businesses/facilities were affected by the 2021 mouse plague. Only those who were affected by the 2021 mouse plague proceeded to the next stage where information about the study was provided. Specifically, the information provided

included an overview of the research being undertaken, privacy, confidentiality, their individual rights as participants, and consent terms.

The collected data was stored in Microsoft Excel and validated by checking for ineligibility, multiple submissions, non-response, and missing values. Substantive validation was done for the numerical entries using systematic observation of descriptive statistics (mean, median, variance, standard deviation, minimum and maximum) in combination with graphical distributions (histograms and box and whisker) to summarize the characteristics of the collected data.

The research was approved by the Commonwealth Scientific and Industrial Research Organisation's Social and Inter-disciplinary Science Human Research Ethics Committee (Ref: 179/22) and aligns with Australia's National Statement on Ethical Conduct in Human Research. The participants, who were 18 years old or over, provided written consent by pressing 'YES' on the online consent form if they agreed to the consent terms. No data that could identify an individual, e.g., name and address were collected. All the participants were de-identified using unique identification codes.

## Data analysis

**Cost analysis framework.** Estimation of the cost of pests such as mice can be done using total cost accounting technique that comprise addition of direct, indirect, and control costs [34]. Direct costs are those that can directly be linked to pests whereas indirect costs are those that cannot. Control costs are costs incurred by the government, e.g., during monitoring of pests. Examples of direct costs include yield losses and reduction in product quality while examples of indirect costs include job losses, property value loss, and loss of market access. However, categorising impacts of pests as either direct, indirect, or control may result in double counting as the distinction between them is not always clear [34]. Furthermore, according to [21] conventional cost analysis offers little guidance in making economic decisions; the authors proposed a loss-expenditure frontier framework an economic efficiency approach for determining avoidable cost, i.e., a combination of losses and expenditures that could have been reasonably be prevented or reduced if a particular activity such as disease control was performed. The avoidable cost can then be used to estimate the level of resources required to manage an outbreak.

The loss-expenditure frontier framework has been used in the past to estimate the economic impact of invasive species in Australia by combining losses with expenditures [35]. The avoidable cost was modified by [36] who only focussed on the avoidable losses (i.e., losses that could have been reasonably be prevented or reduced if pest control was performed) as the key metric for assessing the economic impact of pests on crops. In this study, the loss-expenditure frontier approach was used to define the general cost terms to avoid double counting. Avoidable losses were used as the key metric for comparing the relative efficiency of the chemical mouse bait used to minimize crop yield losses. This enabled the study to offer a standardised metric that can be used for comparison between mouse plagues in the future. The cost of the mouse plague to various entities in this study was viewed as the opportunity cost of prevention, i.e., the cost saving that could be obtained if mouse plagues are prevented from happening. The estimation of the cost of the mouse plague has been described below.

The negative effects of mouse plague from an economic viewpoint inevitably appear either as a loss ($L$), expenditure ($E$), or both, depending on where its impact is felt. For example, a farm enterprise will most likely incur both losses and expenditure whereas households are likely to incur only expenditure. A loss ($L$) implied a benefit that was taken away or a potential benefit that was not realised, e.g., value of crop yield losses due to mouse damage of already planted crops, and value of decreased livestock productivity. The expenditure ($E$) encompassed resources that were allocated to unplanned or non-preferred uses such as buying mouse bait, mouse bait rebates, and repairing or replacing damaged equipment. The value of time spent managing the impact of mouse plague within households, which is a non-monetary cost, was viewed as an opportunity cost of time and was valued using the minimum wage of A\$ 23.2 per hour [37]. To accurately calculate the opportunity cost of time spent cleaning the mess made by mice, it was conservatively assumed that most of the cleaning happened during the peak of the mouse plague which was six months during harvesting of winter cereals;

harvesting of winter cereals mostly occur in autumn and spring in Australia [38]. Using losses ($L$) or expenditure ($E$) as the economic rationale of the cost analysis, the total direct cost ($C$) associated with mouse plague was derived using Equation 1 as part of the loss-expenditure frontier postulated in [21] and [39].

$$C = \sum_{i=1}^{n} (L_i + E_i)$$

(1)

However, the definition of cost as used in the loss-expenditure framework was modified to include damages and value of labour (opportunity cost of time) (see equations 2,3). Specifically, computation of the total cost incurred at the household level ($C_h$) involved valuing the expenditures ($E$) which were incurred by the participants when replacing or repairing damaged household goods and personal vehicles and managing the mice through buying of baits and traps. This also included valuing the labour ($Lk$) spent cleaning after the mess made by the mice, i.e., opportunity cost of time, which is a non-monetary cost to capture the value of labour spent cleaning; [7] found that households spent significant amount of time cleaning after the mess made by mice, but the authors did not quantify the impact. The approach of including value of labour as opportunity cost of time was like the economic valuation done in the transport sector [40]. Additionally, value of damaged goods ($D$) that could not be repaired or replaced was included in the computation of the total cost of the mouse plague to the households. Any mouse bait rebate ($B$) received by the household was deducted from the total amount, as shown in Equation 2, to avoid double counting; affected households, farmers, and businesses were eligible to receive bait rebates from the NSW government [20].

$$C_h = \sum_{i=1}^{n} (E_i + Lk_i + D_i) - \sum_{i=1}^{n} B_i$$

(2)

Calculation of the total cost incurred by the farm enterprise ($C_f$) involved valuing the: 1) expenditures ($E$), 2) value of labour ($Lk$), 3) other damages ($D$) other than crop damages as these were already captured as losses ($L$), hence avoiding double counting, and 4) production losses ($L$) as shown in Equation 3. Computation of the expenditure ($E$) involved valuing the expenditures incurred by the participants when: 1) replacing or repairing damaged farm equipment and vehicles, 2) hiring farm equipment due to damaged equipment, 3) replacing damaged hay or fodder, and 4) managing the mouse plague through buying appropriate mouse baits for use around the farm buildings (e.g., anticoagulant rodenticides) and the paddocks (e.g., zinc phosphide). The value of labour ($Lk$) e.g., spent resowing or replanting damaged crops, and damages ($D$) e.g., value of damaged stock feed, were included when accounting for the total cost of the mouse plague incurred by the farmers. For crop farmers, the losses were the value of planted crops damaged during the mouse plague. The average prices of crops were obtained from [23]. For livestock farmers, the losses were the income forgone due to decrease in livestock productivity as earlier mentioned. Like the computation of the household costs, any bait rebate ($B$) received by the farmer was deducted from the total amount as shown in Equation 3 to avoid double counting. Computation of the total cost incurred by businesses or facilities ($C_b$) was like the farm enterprise calculations with the losses being the reduction in income.

$$C_f = \sum_{i=1}^{n} (E_i + Lk_i + D_i + L_i) - \sum_{i=1}^{n} B_i$$

(3)

The total direct cost of the mouse plague based on the sample collected was obtained by adding the total cost incurred by households, farms, and businesses or facilities. The findings from this study were also compared to the economic impact assessments of past mouse plagues.

The expenditure–loss frontier concept provided a valuable framework for considering the economic impact of the 2021 mouse plague. It enabled us to the capture the total impact of the mouse plague by summing the expenditure on mouse control and production losses. Moreover, it is logical to assume that there is a particular combination of losses and mouse plague control expenditures at which the total economic cost is minimised depending on the density of the mouse plague. Therefore, outlining the impacts of the 2021 mouse plague using expenditure-loss framework enables linkage with other economic damage assessment methods such as economic injury level (i.e., determination of the lowest mouse density that will cause economic damage), economic surplus (an economic framework for measuring welfare change), and Cobb-Douglas production function where mouse bait can be included as additional inputs to assess the overall production behaviour [19,41]. The limitation of the expenditure-loss framework is that it does not capture broader sectoral impacts; sector wide or multisector wide impact analysis was not done in this study due to lack of supply use tables at the state level in Australia.

**Identification of key cost parameters influencing policy instruments.** The entity, i.e., household, farm, and businesses, that was most affected (incurred the highest cost) was selected for further analysis. It was assumed that targeted interventions will be directed towards the most affected entity to significantly reduce the overall impact of mouse plagues in the future. The mouse bait rebate claim scheme was the policy instrument of interest. Consequently, logistic models were developed and analysed using R Statistical Software [42] and marginal effects were used for their interpretation. In the logistic regression model, the expected value of mouse bait rebate claim, assigned $y$, was modelled as shown in Equation 4.

$$E\left[y|x\right] = \frac{1}{1 + e^{-\beta_0 - \beta_1 x}}$$

(4)

Thus, the marginal effect was computed as shown in Equation 5.

$$\frac{\partial E[y|x]}{\partial x} = \beta_1 p(y - 1|x)(1 - p\left(y - 1|x\right))$$

(5)

Where $y$ was the outcome variable, $x$ was the explanatory cost variable, $p$ was the probability of claiming or not claiming mouse bait rebate, $\beta$ was the beta coefficient which represented the change in the log-odds of the outcome of one-unit change in the predictor variable holding all other predictors constant.

### Optimization to compare the relative efficiency of the chemical mouse control options

There are two types of mouse baits namely acute (fast) acting which destroys mice after a few hours and chronic (slow) acting that destroys mice after a few days. During the 2021 mouse plague, zinc phosphide which is an acute rodenticide was used for in-crop baiting whereas anticoagulants which are chronic rodenticides were used around the farm buildings and equipment as these cannot be used near crops due to their high toxicity to humans and are to be used in accordance with regulatory requirements (label conditions) [43]. Consequently, the chemical mouse control options included the use of zinc phosphide, which was used in the fields only, anticoagulant rodenticides which was used against mice around the farm buildings only, combination of zinc phosphide and anticoagulant rodenticides which were both used in the fields and around farm buildings, and the no mouse bait use option.

The optimization modelling of the chemical mouse control options to compare their relative efficiency was done using data envelopment analysis (DEA) framework for crop farmers [44]. The DEA is a powerful tool for modelling efficiency and has been widely used in efficiency optimization studies since its inception in the 1970s [45]. In this study, the chemical mouse control options (excluding no use option as efficiency optimization can't be done mathematically with a unit that has zeros as the input) were viewed as the decision-making units (DMUs).

The efficiency of a DMU was defined as the weighted sum of its outputs divided by the weighted sum of its inputs [46]. The input of interest was the expenditure on a specific type or types of mouse bait as reported by the crop farmers. The output was the avoidable crop yield losses which was the difference between the crop yield losses for each DMU and the control group of participants that didn't use any rodenticide (the no mouse bait use option). In the development of the input-output oriented envelopment model, the main undertaking was to find the combination of the DMUs that produced a target, i.e., an optimal level of inputs and outputs for an inefficient DMU to achieve efficiency. It was assumed that more expenditure on mouse bait by a farmer or a group of farmers or via a mouse bait rebate scheme would result in lower crop yield losses up to a certain point after which mouse control using chemical methods will experience diminishing returns. Using this assumption, the DEA was used to find the optimal level of inputs and outputs as stated earlier. Although chemical methods were the main consideration in this study due to government policy, their efficacy was not considered as the focus of the study was on the economic efficiency of the methods used to control mice during outbreaks. Non-chemical mouse control methods, e.g., electric fencing and biological control were also not considered as they are unlikely to be the main methods of controlling mice during outbreaks at the farm level.

In the envelopment model, the combination of the DMUs was denoted by a vector $\lambda_j$, which was the specific amount of a DMU $j$ used in setting the target for performance for the DMU being evaluated, $k$. Specifically, the maximum possible input (mouse bait expenditure) reduction in $k$'s input, or conversely, the minimum amount of input (mouse bait expenditure) that could be used by the target while producing the same or more of the output (avoidable crop yield losses) was computed using Equation 6.

$$\min \theta$$

$$s.t. \sum_{j=1}^{n^u} x_{i,j}\lambda_j \leq \theta x_{i,k} \forall\, i$$

$$\sum_{j=1}^{n^u} y_{r,j}\lambda_j \geq y_{r,k} \forall\, r$$

$$\lambda_j \geq 0 \ \forall\, j \tag{6}$$

Where $\theta$ was the radial reduction in the amount of DMU $k$'s input (mouse bait expenditure), $n^u$ was the number of DMUs, $\sum_{j=1}^{n^u} x_{i,j}\lambda_j$ referred to the amount of the $i$'th input (mouse bait expenditure) used by the target, and $\sum_{j=1}^{n^u} y_{r,j}\lambda_j$ referred to the $r$'th output (avoidable crop yield losses) produced by the target. The DEA was done using R Statistical Software [42] and the *ompr* r package was used for the algebraic modelling optimization [47].

## Results and discussion

A total sample (*N*) of 1,691 residents in regional NSW participated in this study. Females were the predominant gender comprising 71.14% of the total respondents; 27.79% were male, 0.37% were non-binary and 0.70% preferred not to identify their gender. The average age of the respondents was 48.70 years (standard deviation (SD) = 15.64). Of the 1,691 respondents, 52.34% were random non-panel respondents (completion rate of 10% via this recruitment method) while 47.66% were the pre-recruited panel (completion rate of 10% and a survey dropout rate of 13% via panel recruitment). The total number of farmers was 306 (18.09% of the total number of respondents) of which 158 (51.62%) respondents

were livestock farmers, 27 (8.84%) respondents were crop farmers, and 121 (39.54%) respondents were both crop and livestock farmers.

The total number of responses to the predefined social, economic, and mental (psychological) impacts was 15,069 with excessive cleaning (84.44% of the total responses) being the most mentioned impact as shown in Table 1. Other major impacts included damage to household goods (74.87% of responses), unbearable smell (61.16%), and mental stress (54.10%). Spending considerable time caring for vulnerable family members, poor physical health, and unemployment were the least experienced impacts at the household level as shown in Table 1. In summary, the primary social impact was excessive cleaning whereas damages to household property was the major economic impact of the mouse plague at the household level. Unbearable smell and mental stress were the main mental impacts.

According to the study survey, there were a total of 1,668 (n = 1,668, or 98.6% of respondents) households that were affected by the 2021 mouse plague; 23 (1.4%) households were not affected. The total number of hours spent cleaning because of the mouse plague was 3,982 per day (mean = 2.38; SD = 2.23) equating to A\$ 16,650,334 (mean = 9,846.44) as the total value of time spent cleaning the mess created by the mice during the mouse plague. The total expenditure on cleaning detergents and related items was \$87,331 (mean = 52.26; SD = 74.83) per week equating to A\$ 3,493,240 during the mouse plague. The value of damaged household goods as well as expenditure on vehicle repair and mouse bait was A\$ 6,145,034, A\$ 1,964,798, and A\$ 939,367, respectively.

The total cost (non-monetary and monetary cost) incurred by all the affected households was A\$ 29,192,773 with the non-monetary cost comprising 57.03% (A\$ 16,650,334) of the total cost compared to monetary cost (A\$ 12,542,439) which was 42.97% of the total cost (Table 2). The average monetary ('real') household cost associated with the mouse plague was A\$ 7,519.20 per household (Table 2).

There were 279 responses to keeping of various livestock types with most farmers keeping beef cattle (28.67% of the responses), sheep (22.22% of the responses), and beef cattle and sheep (21.86%). For livestock farmers, the total value of hay and fodder damaged by mice during the mouse plague was A\$ 6,278,286. The total expenditure on additional hay and fodder the livestock farmers had to purchase to replace the damaged ones was A\$ 5,604,570 whereas the total expenditure on other items such as veterinary bills, repair of farm sheds, hiring aeroplane for baiting, traps, fuel cost to buy baits and traps was A\$ 845,805. Therefore, the total expenditure incurred by the livestock farmers was A\$ 6,450,375 with an average of A\$ 23,119.62 per livestock farmer. The total value of damaged stock feed was estimated to be A\$ 3,111,613 resulting in the total value of damaged goods (i.e., damaged hay/fodder and stock feed) to be A\$ 9,389,899 or an average of A\$33,655.55 per livestock farmer. The total value of production losses incurred by the livestock farmers was estimated to be A\$ 4,535,461 averaging A\$ 16,256.13 per livestock farmer. The total cost, i.e., total expenditure,

**Table 1. Frequency of the responses to the structured socio-economic impact assessment.**

| Social, economic, and mental impact | Number of responses (% response) | |
|---|---|---|
| | No | Yes |
| Excessive cleaning | 260 (15.56) | 1,411 (84.44) |
| Damage to household goods | 420 (25.13) | 1,251 (74.87) |
| Unbearable smell | 649 (38.84) | 1,022 (61.16) |
| Mental stress | 797 (45.90) | 904 (54.10) |
| Unsanitary living conditions | 984 (58.89) | 687 (41.11) |
| Damage to household vehicles | 1,168 (69.90) | 503 (30.10) |
| Spending considerable time caring for the elderly or vulnerable family members | 1,535 (91.86) | 136 (8.14) |
| Poor physical health | 1,551 (92.82) | 120 (7.18) |
| Unemployment | 1,658 (99.22) | 13 (0.78) |

**Table 2. Cost of mouse plague at the household level for all the sampled households (n = 1,668).**

| Item | Value (in A$) | Mean |
|---|---|---|
| **Non-monetary cost** | | |
| a) Value of labour | | |
| Value of labour spent cleaning | 16,650,334 | 9,982.20 |
| Subtotal | 16,650,334 | 9,982.20 |
| **Monetary cost** | | |
| b) Damages | | |
| Value of damaged household goods | 6,145,034 | 3,684.00 |
| c) Expenditures | | |
| Expenditure on cleaning supplies | 3,493,240 | 2,094.20 |
| Expenditure on vehicle repair | 1,964,798 | 1,177.90 |
| Expenditure on bait | 939,367 | 563.10 |
| Subtotal | 12,542,439 | 7,519.20 |
| Total | 29,192,773 | 17,501.40 |

total damages, and production losses, incurred by the livestock farmers was A$ 20,375,735 averaging A$73,031.30 per livestock farmer.

Most (66.89%) of the crop farmers had a mixture of more than one broadacre crops as shown in Table 3. This was followed by the crop farmers growing only one type of broadacre crop at 24.32%. The types of crops grown by the crop farmers have been summarised in Table 3. Additionally, the total area of wheat, barley, and oats grown was 55,000 hectares, 24,040 hectares, and 9,752 hectares, respectively. The total area of faba bean, field pea, lentils, lupin, and vetch grown was 6,200, 325, 10, 2,155, and 2,195 hectares, respectively. Canola was grown in 18,414 hectares.

The total area of crops grown for barley, wheat, oats, canola, safflower, legumes, cotton, horticulture, rice, and sorghum were 24,040, 55,000, 9,792, 18,414, 50, 20,611, 2,206, 1,310, 130, and 3,996 hectares, respectively. Therefore, the total area of crops grown was 135,549 hectares with winter cereals comprising 65.53%. The areal extent of the crop area grown (135,549 hectares) as reported by the respondents was equivalent to 8.35% of the total area of crops grown in NSW.

The total expenditure buying seeds for resowing was A$ 1,083,175 averaging A$ 7,318.75 per crop farmer. The total value of damaged seeds was A$ 1,200,553. There were ten farmers whose grains were rejected at the depot and the average percentage of grains rejected at the depot was 37.33% (min = 5%; max = 80%) valued at A$ 172,750. Hence, the total value of damages was A$ 1,373,303 averaging A$ 9,279.07 per crop farmer. The total time spent resowing due to mice damage was 367 hours (mean = 2.62; SD = 4.54) valued at A$ 8,545 with an average of A$ 27.62.

The total value of crop yield losses was A$ 39,558,295 averaging A$ 267,285.77 per crop farmer. The average crop yield loss was 21.82% (minimum: 1%; maximum: 63%). The total crop area damaged was 27,454.57 hectares. Consequently, the impact of the mouse plague was estimated to be A$ 1,440.86 per hectare when only crop yield losses was considered. The total cost, i.e., total expenditure, total damages, and yield losses incurred by the crop farmers was A$ 42,023,318 averaging A$ 283,941.33 per crop farmer.

The total expenditure on repairing damaged farm equipment was A$ 2,844,515 for all the farmers (n = 306) whereas the total expenditure on hiring additional farm equipment was A$ 385,435. The total expenditure on anticoagulant mouse bait for use around farm buildings was A$685,431 for all the farmers and the total expenditure on zinc phosphide for use around the paddocks was A$ 785,288. Therefore, the sum of expenses on mouse bait was A$ 1,470,719 or an average of A$ 4,806.2 per farmer. The total expenditure on buying cleaning items for use around the farm was A$ 420,721. By adding all the expenses, the general farm expenditure was A$ 5,121,390. By adding the general farm expenditure to the total cost

**Table 3. Number of responses for the broad types of crops grown.**

| Broad type of crop grown | Number of responses | % response |
|---|---|---|
| One type of broadacre crop only | 36 | 24.32 |
| Horticulture only | 9 | 6.08 |
| Mixture of more than one broadacre crop only | 99 | 66.89 |
| Mixture of broadacre crops and horticulture only | 4 | 2.70 |
| Total | 148 | 100.00 |

incurred by livestock and crop farmers, the total cost incurred by farmers due to the mouse plague was A$ 67,520,443 averaging A$ 220,655.00 per farmer.

The total number of businesses and facilities that reported that they had been affected by the 2021 mouse plague was 183. Accommodation services were the most affected type of business/facility (25 or 13.66% of the total response). The breakdown of the types of businesses affected is as follows: café (9.29%), rural agricultural suppliers (4.92%), restaurants (4.92%), retail shops (4.37%), mechanic (3.28%), professional services (2.73%), health clinics (2.19%), veterinary clinics (2.19%), bakery (1.64%), supermarket (1.64%), yoga and wellness studios (1.64%), and other, e.g., fuel station, farm produce outlets, pet shop, gift shops etc with only one response each (24.04%). Regarding facilities, recreational facilities were the most affected by the mouse plague (8.74% of the total response) followed by schools (3.28%) and hospitals (2.73%). Other types of facilities that were affected by the mouse plague included wastewater treatment plants, community hall, equestrian facilities, libraries, fire stations, disability centres, caravan parks, community sporting facilities etc (8.74% of the total response).

Those owning or managing businesses and facilities reported that they lost a total of A$ 1,574,297 in income, averaging A$ 8,602.71 per business. The total value of damaged goods was A$ 1,214,150. Businesses and facilities spent a total of A$ 370,550 (mean = 2,024.86), A$ 496,262 (mean = 2,711.81), and A$ 252,220 (mean = 1,378.25) on hiring cleaning services, buying bait, and repairing damaged equipment respectively; total expenditure was A$ 1,119,032. Therefore, the total cost incurred by the businesses/facilities was A$ 3,907,479 averaging A$ 21,352.34 per business/facility.

The total expenditures incurred by households, farms, and businesses/facilities was A$ 20,171,377 whereas the total value of the damages was A$ 18,122,386. The total value of time spent cleaning and resowing was A$ 16,658,879 and the total livestock production, crop yield losses, and business income losses was A$ 45,668,053. Consequently, the total direct cost of the mouse plague was A$ 100,620,695 among the sampled population (N = 1,691). The distribution total cost according to the distinct levels (i.e., household, farm, or business/facility) and types of cost (i.e., expenditure, opportunity cost of time, damages, and losses) is shown in Fig 2 and the spatial distribution of the total cost is shown in Fig 3. All relevant data are within the manuscript and its Supporting Information files (S1 File).

As shown in Fig 2, crop yield losses were the most incurred type of cost whereas non-monetary cost was the most incurred type of cost by the households. Spatial distribution of the distinct types of cost indicated that majority of the 2021 mouse plague impact occurred in the central west and northwest regions of NSW; Mudgee, Tamworth, and Walgett were some of the most affected local government areas as shown in Fig 3.

The farmer (crops and livestock) incurred cost in this study was compared with the 1993 mouse plague to assess whether there have been any changes on how the mouse plagues are affecting them [7]. The cost of the 1993 mouse plague on businesses/facilities was excluded as the [7] estimates were averages per business/facility type, but it was not clear how they were derived.

The household costs and the value of rejected grain at the depot were excluded as these were not computed by [7]. Explicitly, the key parameters used to compute the cost of mouse plagues, that is, losses (crop yield losses and livestock production losses), damages, value of labour resowing, and mouse bait expenditures were compared between the current

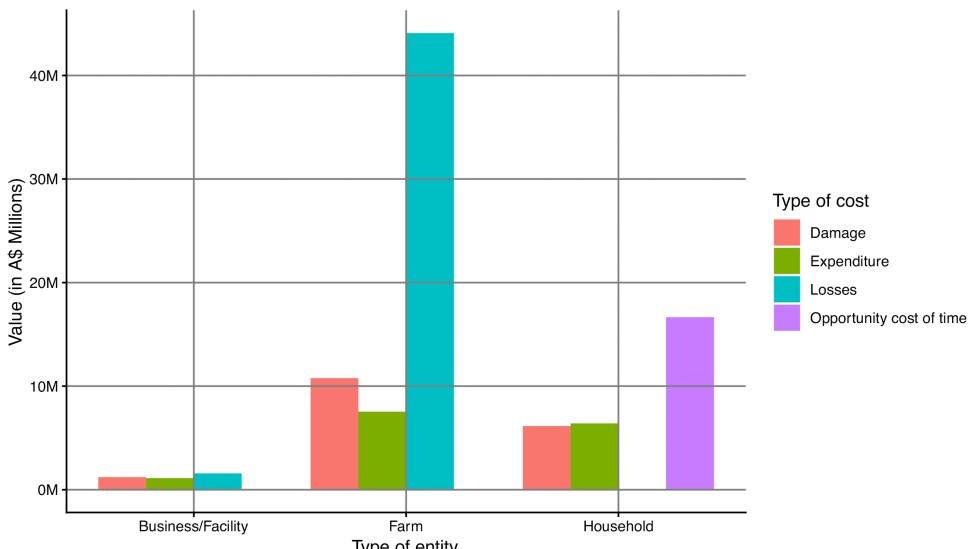

**Fig 2. Distribution of the cost categories between among the affected entities (businesses/facilities, farms, and households).**

study and the 1993 mouse plague. It is worth noting that although our study included broader aspects of expenditures at the farm level, such as, monies spent buying cleaning items, replacing damaged hay/fodder, and hiring additional farm equipment, these were excluded in the comparative analysis as they were excluded in [7]. Table 4 provides a summary of the cost incurred by farmers in this study and [7]. Additionally, Fig 4 shows the cost of mouse plague incurred by the farmers in each Australian state affected in 1980, 1984, and 1993 as reported by [7] compared with the current study.

A comparison of the marginal effects on claiming bait rebate for selected cost parameters was done using average marginal effects. The selected cost parameters included the value of hay/fodder damaged by the mice, bait expenditures, and crop yield loss. These parameters represented the damage, expenditure, and losses components of the total direct cost of the mouse plague as incurred by the farmers who were the most affected entity in the cost analysis.

The average marginal effects for the value of hay/fodder damaged was 0.01453 which meant that on average a one-point increase in the value of hay/fodder damaged was associated with a 1.453 percentage increase in the probability of the affected farmer claiming mouse bait rebate from the NSW government (Table 5). On the contrary, mouse bait expenditures and crop yield losses were associated with a −0.846 and −0.193 percentage point decrease, respectively, as shown in Table 5.

A constant return to scale model was used to compare the efficiency of controlling mouse using anticoagulants, zinc phosphide, and both anticoagulants and zinc phosphide by crop farmers (the most affected type of farmers during the cost analysis). Mouse bait expenditure and avoidable crop yield losses were used as the input and output, respectively. The average mouse bait expenditure for the anticoagulants, zinc phosphide, and both anticoagulants and zinc phosphide group of crop farmers was A\$ 432,670, A\$ 404,352, and A\$ 412,834, respectively. The avoidable losses associated with the anticoagulants, zinc phosphide, and both anticoagulants and zinc phosphide group of crop farmers was A\$ 1,997,915, A\$ 3,955,830, and A\$ 5,142,578, respectively.

The constant returns to scale input-output model for the crop farmers that used anticoagulant, zinc phosphide, and both anticoagulant and zinc phosphide was 0.36, 0.78, and 1.00, respectively. The lambda ($\lambda$) which reflects the way that a best target is made for that unit for the anticoagulant, zinc phosphide, and both anticoagulant and zinc phosphide group was 0.38, 0.76, and 1.00, respectively. These results showed that use of both anticoagulants and zinc phosphide was

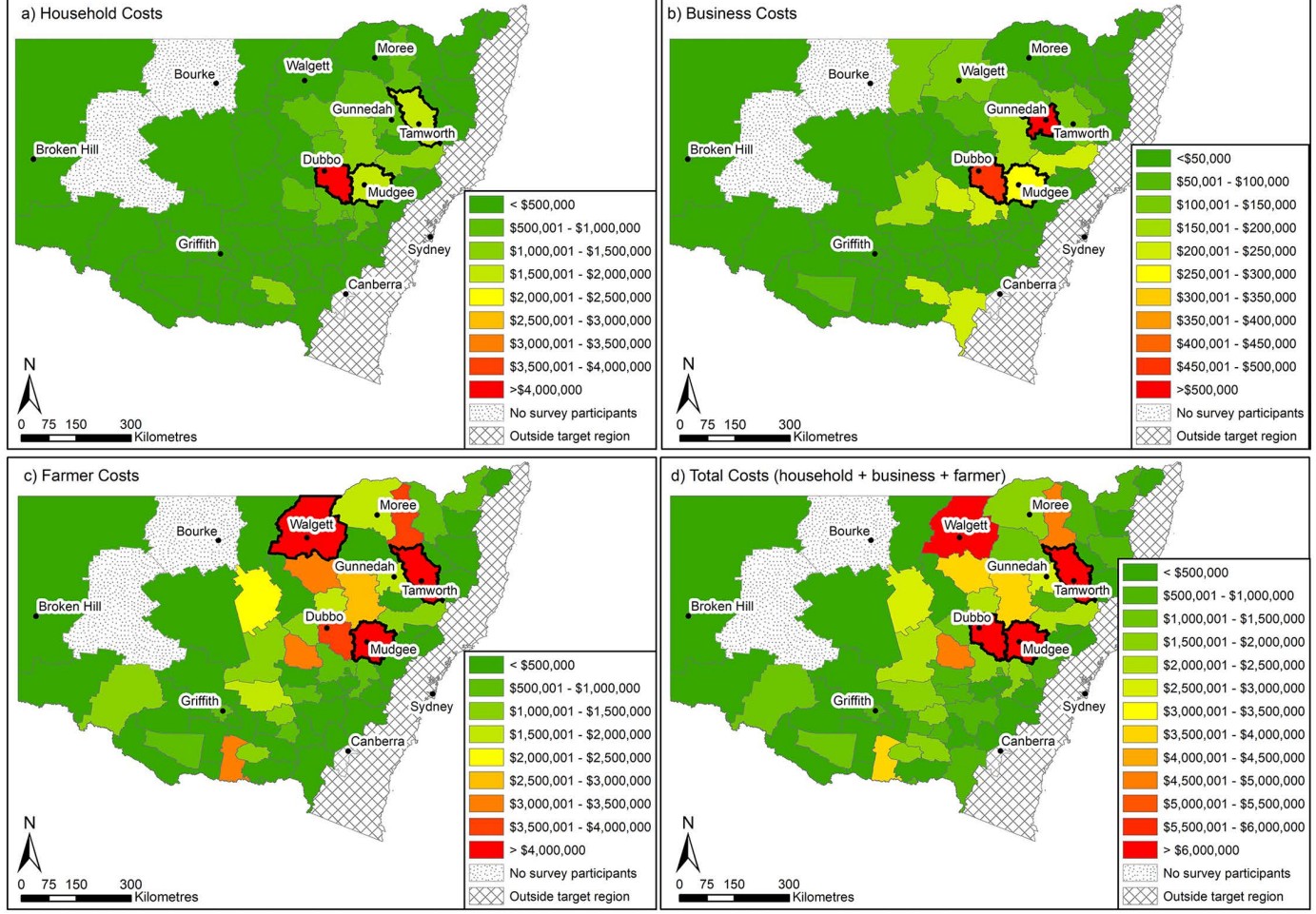

**Fig 3. Spatial distribution of the costs.** Map prepared using ArcGIS Desktop (ArcMap version 10.8.2), Environmental Systems Research Institute (ESRI), Redlands, California, USA. LGA boundaries and town point data incorporate data from the 2021 Administrative Boundaries © Geoscape Australia licensed by the Commonwealth of Australia under Creative Commons Attribution 4.0 International Licence (CC BY 4.0) [48].

**Table 4. Comparison of the cost of mouse plague to farmers in 1993 and 2021.**

| Year (study) | State | Farmer incurred cost in A$ (% of the total cost) | | | | | Total cost (in A$) |
|---|---|---|---|---|---|---|---|
| | | Mouse bait | Value of labour spent resowing | Crop damage (Crop yield losses) | Other damages | Livestock production losses | |
| 1993 [7] | Victoria | 196,100 (1.29) | 748,700 (4.93) | 13,100,000 (86.21) | 350,000 (2.30) | 800,000 (5.26) | 15,194,800 |
| | South Australia | 1,530,500 (3.09) | 910,000 (1.84) | 44,947,500 (90.86) | 1,280,000 (2.59) | 800,000 (1.62) | 49,468,000 |
| 2021 (this study) | New South Wales | 1,470,719 (2.44) | 1,083,175 (1.80) | 39,558,295 (65.65) | 13,607,717 (22.58) | 4,535,461 (7.53) | 60,255,367* |

*Excludes expenditure on replacing damaged hay/fodder, cleaning items, additional hiring of farm equipment, and others such as veterinary bills.

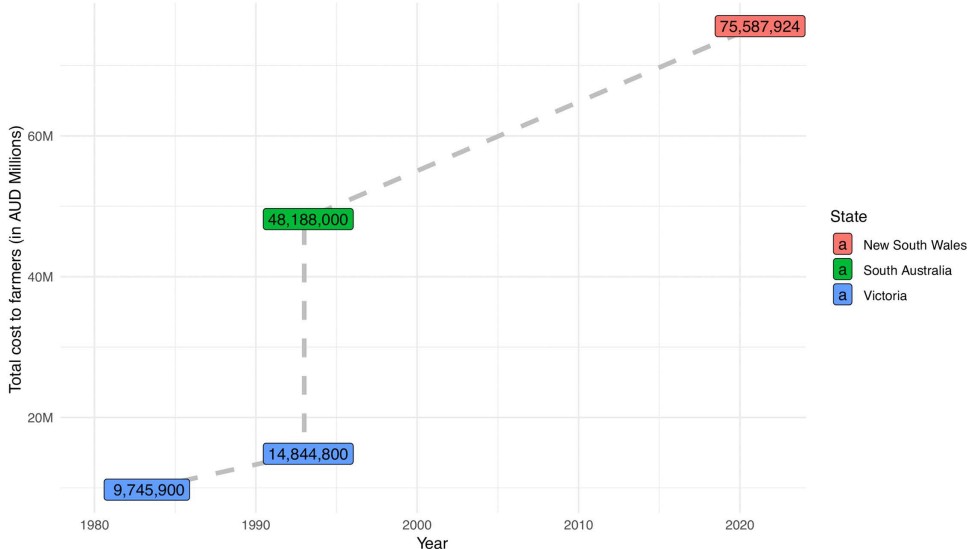

**Fig 4. Cost of mouse plague incurred by the farmers in each Australian State affected in 1980, 1984, and 1993 compared to the 2021 mouse plague in NSW (this study).**

**Table 5. Comparison of the average marginal effects among selected cost parameters.**

| Term | Estimate | Standard error | z | Pr(>\|z\|) | 95% confidence interval | |
|---|---|---|---|---|---|---|
| | | | | | 2.5% | 97.5% |
| Value of hay/fodder damaged | 0.01453 | 0.00616 | 2.35 | 0.0187 | −0.00254 | 0.0316 |
| Mouse bait expenditures | −0.00846 | 0.000265 | −31.92 | 0.0000 | −0.00795 | −0.0897 |
| Crop yield loss | −0.00193 | 0.000507 | −46.9 | 0.0014 | −0.00239 | −0.00147 |

relatively efficient in reducing crop yield losses compared to each being used independently. Use of anticoagulants alone was relatively the least efficient in reducing crop yield losses.

Knowledge of the economic impacts of mouse plagues is limited unlike that of other pests [49–54]. To the best of our knowledge there has been no peer reviewed study on the economic impact of mouse plague in NSW let alone in Australia. This study not only addresses this knowledge gap but also identified some of the key cost parameters that influenced key mouse plague control policy instruments such as the mouse bait rebate scheme. Our study went further to compare the efficiency of chemical mouse bait control methods used during the 2021 mouse plague in reducing crop yield losses. Using this efficiency benchmarking approach as well as determining the key factors that influenced mouse bait claims, the study contributes to the knowledge gap on efficient management of mouse plagues and the methodological gap on how cost data can be used for decision making. The analytical approach used in this study can be applied to other research areas such disaster and invasive pest impact analysis.

Our study found that the total direct cost of the 2021 mouse plague from the survey respondents was A\$ 100,620,695 with the total cost to farmers being A\$ 67,520,443 (67% of the total direct cost). Crop yield losses contributed the most to the total cost of the 2021 mouse plague (39.3% of the total direct cost) and to the total cost incurred by crop farmers (94.1% of the total cost incurred by crop farmers). Our finding on the total direct cost of the mouse plague is comparatively higher than that reported by [7] who reported that the total cost of the 1993 mouse plague was A\$ 64.5 million with farmers incurring most of the cost (93.9% of the total direct cost). A possible explanation is that our study accounted for

the value of time households spent cleaning after the mess made by the mice; this was not accounted for by [7]. Additionally, [7] did not include the value of damaged hay or fodder as was done in this study which can be substantial as reported by [16]. Our study found that crop farmers lost A$ 283,941.3 per farmer compared with A$73,031.3 per livestock farmer reported by [7]; the A$ 283,941.3 loss equates to a farm cash income reduction of 28.3% for each farmer on average.

Still on the comparison between this study and [7], the average crop yield loss per hectare and the average total cost of the 2021 mouse plague to crop farmers was A$ 1,440.86 per hectare and A$ 283,941.3 per crop farmer (excluding general farm costs) compared to A$ 231 per hectare (equivalent to A$ 452.8 in 2021 [55]) and A$ 50,626 per crop farmer (equivalent to A$ 99,236.9 in 2021 [55]) as reported by [7]. The possible explanation for this difference could be due to the increased value of crops in Australia over the years. Other explanations include the differences in heterogeneity between and within farms between the two studies and the potential impact of COVID-19 on prices and availability of labour and materials. Nonetheless, the higher impact of the mouse plague in each affected area as reported in this study may indicate that the economic impact of the mouse plague is becoming more localised and intense due to intensive farming practices over the past 30 years.

A similarity between the present study and that of [7] is that both analyses found that crop yield loss was the most significant contributor to the overall cost of the mouse plague. However, in our study we found that other types of damages (e.g., damaged farm equipment) were higher (22.58% of the total cost of the mouse incurred by the farmers) compared to the [7] findings (2.30% in Victoria and 2.59% in South Australia). Mapping of the economic impact of the 2021 plague revealed that the central west and northwest regions of NSW were the most affected regions. This finding was comparable to [2] who reported that the 2021 mouse plague mostly occurred in the central west and northwest regions of NSW. Besides, both crop and livestock farmers were equally affected. For example, broadacre crop farmers (i.e., those producing large-scale grains, and oilseeds), and horticultural farmers lost most of their crops due to the damage caused by mice at the early and late stages of planting as indicated by the crop yield losses in this study. They also lost some of their stored seeds for planting. Livestock farmers lost most of their livestock feed such as hay and fodder because of the damage caused by the mice; mice usually use hay or fodder as breeding ground. Further, livestock farmers incurred production losses potentially due to the stress caused by the mice on their animals and lack of feed because of the damaged haystacks and or fodder.

The economic impacts of the 2021 mouse plague can be deemed severe as it reduced the crop yield by approximately 21.82% as shown in this study. The severity of the economic impact is best reflected by extrapolating the study findings particularly the crop yield losses to the entire state of NSW. However, this was a challenging undertaking due to limited information on the total affected area of the crops grown across the entire state. The 2021 mouse plague has been estimated to cost over A$ 1 billion according to NSW Farmers (a farmer advocacy association) [20]. Unfortunately, there were no details on how the amount was derived and so further comparisons could not be discussed.

This study found that the economic impact of the mouse plague depends on the type of premises with farm enterprises being affected the most. The economic impact also varied between and within the premises. For example, damages were the main economic impact of the mouse plague among livestock farmers compared to crop farmers; A$ 9,389,899 or A$33,655.5 per livestock farmer compared to A$ 1,373,303 per crop farmer or an average of A$ 9,279. The high cost of the damages incurred by the livestock farmers was potentially due to the considerable number of livestock producers keeping beef cattle that required supplementary feeding with hay or fodder after the subsequent drought. However, production losses were higher among crop farmers compared to livestock farmers; averaging A$ 283,941.3 per crop farmer compared to an average of A$ 16,256 for the livestock farmers. Equally, intangible costs such as value of labour spent cleaning was comparatively higher in households. This was also confirmed by the high frequency of responses to excessive cleaning which was 84% of the responses to the socio-economic impact; studies done by [56,57] indicated that the psychological and social impacts of the 2021 mouse plague was substantial.

Analysis of the cost parameters that influenced the mouse bait rebate scheme revealed that the value of damaged hay or fodder increased the likelihood of the affected farmer claiming bait rebate. This may indicate that most farmers claiming mouse bait rebate are livestock farmers. On the contrary, crop yield loss negatively influenced mouse bait reclaim indicating that crop farmers are less likely to seek mouse bait rebate. The plausible reason for this was that the magnitude of the crop yield losses compared with the amount farmers could claim as mouse bait rebate was substantial making affected farmers not to seek the mouse bait rebate. Our study also found that bait expenditure negatively influenced the likelihood of a farmer seeking mouse bait rebate. The possible explanation was that the bait expenditures well exceeded the amount of mouse bait rebate received by the farmers. Furthermore, zinc phosphide usually requires wider application using aircraft significantly increasing the total cost incurred by crop farmers when using chemical control methods to manage mouse plagues. Overall, it can be deduced that the type of farming influenced the likelihood of farmers claiming or not claiming mouse bait rebate. However, more research is required to understand the barriers and effectiveness of the mouse bait rebate scheme as well as other factors influencing its use.

This study went further to understand the relative efficiency of the mouse control options which were mostly based on the type of mouse bait used. Our study found that the efficiency of the chemical mouse bait in reducing crop yield losses was dependent on the type or types of mouse bait used. For example, our study revealed that the combination of anticoagulants and zinc phosphide was relatively more efficient in reducing crop yield losses than each type of mouse bait used independently; combination of anticoagulants and zinc phosphide was able to produce the same level of output (i.e., avoidable losses) with 64% and 22% less input (mouse expenditure) if each was used independently. This was not surprising as use of both anticoagulants (around farm equipment and haystacks) and zinc phosphide (in crops) will most likely result in substantial reduction of mouse densities and ultimately in minimal avoidable crop yield losses. Practically, achievement of high efficiency of mouse control using chemical mouse baits can be done by encouraging crop farmers to use both mouse baits that can destroy mice within a short and prolonged period. Farmers can also be encouraged to practice broadacre application of mouse bait before sowing. However, more research is required to understand the efficiency of all mouse control methods (both chemical and non-chemical) available to farmers as well as the possible set of variables influencing farmers choice of these methods, e.g., human beliefs and preferences. Additionally, apart from conducting further research, other preventive measures such as increased surveillance, farmer awareness on actions to take before and during the mouse plague as well as deep planting of seeds can be implemented to reduce the impact of the mouse plague [16,58,59].

It is worth mentioning that farmers restricted their use of anticoagulants to around buildings and zinc phosphide to pastures because of NSW government safety regulation requirements and recommendations as they can have negative health impacts on the users (e.g., farmers handling the mouse bait), other people, animals, and the environment. Besides, unlike anticoagulants, zinc phosphide works best in pastures, cropping, and stubble. It also degrades easily in the environment (hence reduced chance of accidental poisoning of other species) and can efficiently be applied using both ground-based methods such as spreaders and aerially [60]. On the contrary, anticoagulants are known to be poisonous to a wide range of species including mammals, fish, and birds and are less effective in wet environments [61].

While this study provides a detailed analysis of the economic costs emerging from the 2021 mouse plague, it had some limitations. First, empirical damage assessments particularly on crops to establish damage thresholds were not done as the economic analysis was carried out after the mouse plague. However, this was unavoidable due to the time lag in acquiring resources for the economic analysis. Second, the study was conducted two years after the mouse plague, and this may result in under or over-estimation of certain parameters. Regardless, the study ensured that bias was limited using probabilistic sampling and careful designing of the survey instrument to reduce mental effort for the respondents [62]. This resulted in the research results being comparable with past studies. Third, the study most likely underestimated the total economic of the mouse plague as wider economic impacts were not captured. Despite these limitations, this study provides a robust economic approach that enables the impact of mouse plagues to be compared with other disasters that affect households, farmers, and businesses or facilities such as floods and droughts.

## Conclusions

Our study empirically assessed the broader impact of the 2021 mouse plague in NSW reducing the knowledge gap on the economic impact of mouse plagues. The study found that the economic impact of mouse plague is substantial with farmers bearing most of the cost especially due to crop yield losses. Households were equally affected by the mouse plague as they spent considerable household labour and time cleaning after the mess made by mice. Considering the high farm production losses, there is need to strengthen support for farmers as well as initiate or strengthen preventive measures against future mouse plagues. This can partly be achieved by strengthening policy instruments such as the mouse bait rebate schemes. Equally, the study recommends use of both slow and fast acting mouse baits to minimise crop yield losses but further research investment on the efficiency of all mouse control options available to farmers as well as the factors that influence use of mouse bait may be required.

## Supporting information

**S1 File. Data used in the analysis.**
(XLSX)

## Acknowledgments

The authors acknowledge the NSW community for participating in the study and providing the data that underpinned this research. We also thank the online research provider for assisting in data collection, Dr. Loechel Barton for evaluating the questionnaire, and various community groups and research facilitators for distributing the survey link.

## Author contributions

**Conceptualization:** Walter Okello, Peter R. Brown.

**Data curation:** Walter Okello, Kerry Collins.

**Formal analysis:** Walter Okello, Aditi Mankad, Lucy Carter.

**Funding acquisition:** Peter R. Brown.

**Investigation:** Walter Okello, Kerry Collins, Aditi Mankad, Lucy Carter.

**Methodology:** Walter Okello.

**Project administration:** Walter Okello, Kerry Collins.

**Validation:** Walter Okello, Kerry Collins, Aditi Mankad, Lucy Carter, Peter R. Brown.

**Visualization:** Walter Okello, Kerry Collins.

**Writing – original draft:** Walter Okello, Kerry Collins, Aditi Mankad, Lucy Carter, Peter R. Brown.

**Writing – review & editing:** Walter Okello, Kerry Collins, Aditi Mankad, Lucy Carter, Peter R. Brown.

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
