## [Decision Letter · Decision Letter 0]

5 Jan 2026

PONE-D-25-61238Multi-analysis of the 2021 mouse plague economic impact in New South Wales, Australia: Quantifying cost, policy variables, and efficient control optionsPLOS One

Dear Dr. Okello,

Thank you for submitting your manuscript to PLOS ONE. After careful consideration, we feel that it has merit but does not fully meet PLOS ONE’s publication criteria as it currently stands. Therefore, we invite you to submit a revised version of the manuscript that addresses the points raised during the review process.

We look forward to receiving your revised manuscript.

Kind regards,

Timothy Omara

Academic Editor

PLOS One

Journal Requirements:

The study was funded by the Department of Regional NSW “Research and monitoring

program to support more effective management of mice in NSW” with support from

CSIRO Health & Biosecurity.

The authors acknowledge the NSW community for participating in the study and providing the data that underpinned this research. The authors acknowledge NSW Department of Primary Industries and the Grains Research & Development Corporation for their guidance and support. We also thank the online research provider for assisting in data collection, Dr. Loechel Barton for evaluating the questionnaire, and various community groups and research facilitators for distributing the survey link.

The study was funded by the Department of Regional NSW “Research and monitoring

program to support more effective management of mice in NSW” with support from

CSIRO Health & Biosecurity.

7. Please amend your list of authors on the manuscript to ensure that each author is linked to an affiliation. Authors’ affiliations should reflect the institution where the work was done (if authors moved subsequently, you can also list the new affiliation stating “current affiliation:….” as necessary).

8. We note that Figure 3 in your submission contain [map/satellite] images which may be copyrighted. All PLOS content is published under the Creative Commons Attribution License (CC BY 4.0), which means that the manuscript, images, and Supporting Information files will be freely available online, and any third party is permitted to access, download, copy, distribute, and use these materials in any way, even commercially, with proper attribution. For these reasons, we cannot publish previously copyrighted maps or satellite images created using proprietary data, such as Google software (Google Maps, Street View, and Earth). For more information, see our copyright guidelines: http://journals.plos.org/plosone/s/licenses-and-copyright.

a. You may seek permission from the original copyright holder of Figure 3 to publish the content specifically under the CC BY 4.0 license.

We recommend that you contact the original copyright holder with the Content Permission Form (httpp://journals.plos.org/plosone/s/file?id=7c09/content-permission-form.pdf) and the following text:

Additional Editor Comments:

Dear authors,

Thank you for resubmitting your manuscript to PLOS ONE. Reviewers have recommended reconsideration of the draft following minor revisions. Please, consider the following during revision.

1. The current title is repetitive. A strong title usually answers one MAIN QUESTION, and rarely three. That is:

(a) ‘Economic impact’ already implies costs

(b) ‘Quantifying cost’ restates that same idea

(c) The ‘policy variables’ and ‘control options’ to my understanding relates to intervention and/or management.

Two non-repetitive alternative titles I would suggest are:

A multi-dimensional assessment of the 2021 mouse plague in New South Wales, Australia: Economic impacts and policy responses

Economic impacts and control strategies of the 2021 mouse plague in New South Wales, Australia

2. Please format the intext citations and references to the journal style.

3. I have added some comments and suggestions in the attached MS file which could improve manuscript clarity.

Reviewers' comments:

Reviewer's Responses to Questions

**Comments to the Author**

1. Is the manuscript technically sound, and do the data support the conclusions?

Reviewer #1: Yes

Reviewer #2: Yes

2. Has the statistical analysis been performed appropriately and rigorously? 

Reviewer #1: Yes

Reviewer #2: Yes

3. Have the authors made all data underlying the findings in their manuscript fully available?

The PLOS Data policy requires authors to make all data underlying the findings described in their manuscript fully available without restriction, with rare exception (please refer to the Data Availability Statement in the manuscript PDF file). The data should be provided as part of the manuscript or its supporting information, or deposited to a public repository. For example, in addition to summary statistics, the data points behind means, medians and variance measures should be available. If there are restrictions on publicly sharing data—e.g. participant privacy or use of data from a third party—those must be specified.requires authors to make all data underlying the findings described in their manuscript fully available without restriction, with rare exception (please refer to the Data Availability Statement in the manuscript PDF file). The data should be provided as part of the manuscript or its supporting information, or deposited to a public repository. For example, in addition to summary statistics, the data points behind means, medians and variance measures should be available. If there are restrictions on publicly sharing data—e.g. participant privacy or use of data from a third party—those must be specified.requires authors to make all data underlying the findings described in their manuscript fully available without restriction, with rare exception (please refer to the Data Availability Statement in the manuscript PDF file). The data should be provided as part of the manuscript or its supporting information, or deposited to a public repository. For example, in addition to summary statistics, the data points behind means, medians and variance measures should be available. If there are restrictions on publicly sharing data—e.g. participant privacy or use of data from a third party—those must be specified.requires authors to make all data underlying the findings described in their manuscript fully available without restriction, with rare exception (please refer to the Data Availability Statement in the manuscript PDF file). The data should be provided as part of the manuscript or its supporting information, or deposited to a public repository. For example, in addition to summary statistics, the data points behind means, medians and variance measures should be available. If there are restrictions on publicly sharing data—e.g. participant privacy or use of data from a third party—those must be specified.

Reviewer #1: Yes

Reviewer #2: Yes

4. Is the manuscript presented in an intelligible fashion and written in standard English?

Reviewer #1: Yes

Reviewer #2: Yes

5. Review Comments to the Author

Reviewer #1: Well thought out and systematic analysis on the economic impact of mouse plague with findings, potential limitations clearly aritculated and appropriate implications for policy and practice outlined. A pleasure to read -- congratulations!

Reviewer #2: The manuscript titled ‘Multi-analysis of the 2021 mouse plague economic impact in New South Wales, Australia: Quantifying cost, policy variables, and efficient control options’ is interesting and need of the hour as it focuses on the agrarian impact of the 2021 mouse plague. It is disturbing to note that even after being severely impacted by the mouse plague, farmers themselves had to bear 67.10% of the total cost of loss. This result forms a valuable insight, seeking planned interventions from the policy front.

1. Authors are advised to revise the figures and graphs, especially figure. 3, as figure texts are not legible.

2. Please mention how all the farmers including cereals, pulses, vegetables, livestock etc are equally impacted by the mouse plague? sampling gave equal weightage to all farmers alike? Is it the ground reality .

3. Kindly mention in the discussion section, why farmers were restricting/ limiting mouse control options as only the chemical used by crop farmers to reduce crop yield losses, and have they found it to be counterproductive or any negative impacts of it health of the produce or farmers.

4. grammatical/ typo errors have to be corrected.

6. PLOS authors have the option to publish the peer review history of their article (what does this mean?). If published, this will include your full peer review and any attached files.). If published, this will include your full peer review and any attached files.). If published, this will include your full peer review and any attached files.). If published, this will include your full peer review and any attached files.

...

Reviewer #1: No

Reviewer #2: **Yes:** Dhanya PunnoliDhanya PunnoliDhanya PunnoliDhanya Punnoli

---

## [Author Response · Author response to Decision Letter 1]

23 Mar 2026

We have now revised the manuscript according to the reviewers comments. Please see the revised manuscript (submitted as 'Manuscript' and 'Revised Manuscript with Track Changes') as well as our response to the reviewers comments. We have now revised the figures and used NAAS tool to ensure that they conform to the journal requirements. Fig 3 is much more legible as TIFF especially using zooming. All the various types of the farmers were equally impacted by the mouse plague, and this is the reality on the ground. We have now highlighted this in the results and discussion section (L569-576). The restriction of the use of the chemical options was because of the government safety regulation requirements and recommendations as they can have negative health impacts on users (e.g., farmers handling the mouse bait), other people, animals, and the environment. We have now highlighted this in the results and discussion section (L632-640). We have corrected the typo errors. For example, we have revised ‘plaque’ to ‘plague’ (L100), and ‘Accommodation services was’ to ‘Accommodation services were’ (L443).

---

## [Editor Report · Decision Letter 1]

24 Mar 2026

A multi-dimensional assessment of the 2021 mouse plague in New South Wales, Australia: Economic impacts and policy responses

PONE-D-25-61238R1

Dear Dr. Okello,

We’re pleased to inform you that your manuscript has been judged scientifically suitable for publication and will be formally accepted for publication once it meets all outstanding technical requirements.

Kind regards,

Timothy Omara

Academic Editor

PLOS One
---

## [Editor Report · Acceptance letter]

PONE-D-25-61238R1

PLOS One

Dear Dr. Okello,

I'm pleased to inform you that your manuscript has been deemed suitable for publication in PLOS One. Congratulations! Your manuscript is now being handed over to our production team.

Kind regards,

on behalf of

Dr. Timothy Omara

Academic Editor

PLOS One